# Differentially Private Gradient Boosting on Linear Learners for Tabular Data Analysis

**Saeyoung Rho** [*]
Columbia University
s.rho@columbia.edu

**Cedric Archambeau**
Amazon AWS AI/ML
cedrica@amazon.com

**Sergul Aydore**
Amazon AWS AI/ML
saydore@amazon.com

**Beyza Ermis**
Amazon AWS AI/ML
ermibeyz@amazon.com

**Michael Kearns**
Amazon AWS AI/ML
University of Pennsylvania
mkearns@cis.upenn.edu

**Aaron Roth**
Amazon AWS AI/ML
University of Pennsylvania
aaroth@cis.upenn.edu

**Shuai Tang**
Amazon AWS AI/ML
shuat@amazon.com

**Yu-Xiang Wang**
Amazon AWS AI/ML
UC Santa Barbara
yuxiangw@cs.ucsb.edu

**Zhiwei Steven Wu**
Amazon AWS AI/ML
Carnegie Mellon University
zhiweiw@andrew.cmu.edu

## Abstract

Gradient boosting takes *linear* combinations of weak base learners. Therefore, absent privacy constraints (when we can exactly optimize over the base models) it is not effective when run over base learner classes that are closed under linear combinations (e.g. linear models). As a result, gradient boosting is typically implemented with tree base learners (e.g., XGBoost), and this has become the state of the art approach in tabular data analysis. Prior work on private gradient boosting focused on taking the state of the art algorithm in the non-private regime—boosting on trees—and making it differentially private. Surprisingly, we find that when we use differentially private learners, gradient boosting over trees is not as effective as gradient boosting over linear learners. In this paper, we propose differentially private gradient-boosted linear models as a private classification method for tabular data. We empirically demonstrate that, under strict privacy constraints, it yields higher F1 scores than the private versions of gradient-boosted trees on five real-world binary classification problems. This work adds to the growing picture that the most effective learning methods under differential privacy may be quite different from the most effective learning methods without privacy.

## 1 Introduction

Gradient boosting is an approach to learn an additive model such that the sum of many weak base learners approximates the final output [1]. This is achieved by iteratively fitting the next base learner to the gradient of the loss evaluated at the current prediction. Algorithm 1 outlines gradient boosting in a general form, which can be parameterized by any choice of loss function $\mathcal{L}$ and base learner $b(x)$. Classification and regression trees (CARTs) are one of the most popular choices for the base learner because of its effectiveness on tabular data and deployment-ready tree-based data structures

---

[*]Saeyoung is the lead author; all other authors are listed in alphabetical order. Saeyoung performed this work during an internship at AWS AI/ML

2022 Trustworthy and Socially Responsible Machine Learning (TSRML 2022) co-located with NeurIPS 2022.

in systems. Existing packages, including XGBoost [2], LightGBM [3], and CatBoost [4], drastically improved the usability of gradient boosting among tools for tabular data analysis.

---

**Algorithm 1** Gradient Boosting (iterations $T$, loss $\mathcal{L}$, base learner $b(x; \theta)$)

---

 **Data input:** covariates $x_1, \cdots, x_n$ and labels $y_1, \cdots, y_n$.
 **Initialize** $f(x) = 0$
 **for** $t \in [T]$ **do**
   Compute $\theta_t = \arg\min_\theta \sum_{i=1}^n \mathcal{L}(y_i, f(x_i) + b(x_i; \theta))$
   Update $f(x) \leftarrow f(x) + f_t(x)$, where $f_t(x) = b(x; \theta_t)$
 **end for**
 **return** $f(x) = \sum_{t=1}^T f_t(x)$

---

There has been an increasing demand for privacy-preserving machine learning tools, which naturally triggered a wave of efforts to develop a private version of the gradient boosting algorithm. Differential privacy (DP, Definition 1.1) is one of the most prevalent definitions of privacy, and was adopted to make gradient boosting algorithms private in recent works [5, 6, 7]. DP ensures that, for a randomized algorithm, when two neighboring datasets that differ in one data point are fed into an algorithm, the two outputs are indistinguishable, within some probability margin defined using $\epsilon$ and $\delta \in [0, 1)$.

**Definition 1.1** (Differential Privacy [8]). *A randomized algorithm $\mathcal{M}$ with domain $\mathcal{D}$ is $(\epsilon, \delta)$-differentially private for all $\mathcal{S} \subseteq Range(\mathcal{M})$ and for all pairs of neighboring databases $D, D' \in \mathcal{D}$,*

$$\Pr[\mathcal{M}(D) \in \mathcal{S}] \leq e^\epsilon \Pr[\mathcal{M}(D') \in \mathcal{S}] + \delta, \tag{1}$$

*where the probability space is over the randomness of the mechanism $\mathcal{M}$.*

As an extension of this idea, a single-parameter family of privacy notion (Gaussian differential privacy, GDP) was later proposed [9]. We first define the trade-off function $T(P, Q)$ and use it to define GDP.

**Definition 1.2** (Trade-off function, Definition 2.1 of [9]). *For any two probability distributions $P$ and $Q$ on the same space, the trade-off function $T(P, Q) : [0, 1] \to [0, 1]$ is defined as*

$$T(P, Q)(\alpha) = \inf_\phi \{1 - \mathbb{E}_Q[\phi] : \mathbb{E}_P[\phi] \leq \alpha\}$$

**Definition 1.3** (Gaussian Differential Privacy, Definition 2.6 of [9]). *A mechanism $\mathcal{M}$ is said to satisfy $\mu$-Gaussian Differential Privacy ($\mu$-GDP) if it is $G_\mu$-DP. That is,*

$$T(\mathcal{M}(D), \mathcal{M}(D')) \geq G_\mu$$

*for all neighboring datasets $D$ and $D'$, where $G_\mu = T(\mathcal{N}(0, 1), \mathcal{N}(\mu, 1))$.*

$\mu$-GDP means that determining whether an individual's data is present in the dataset from one draw is at least as difficult as telling apart the two normal distributions $\mathcal{N}(0, 1)$ and $\mathcal{N}(\mu, 1)$. $\mu$-GDP can be converted to $(\epsilon, \delta)$-DP and vice versa.

**Corollary 1.1** (Conversion between GDP and DP, Corollary 2.13 of [9]). *A mechanism is $\mu$-GDP if and only if it is $(\epsilon, \delta(\epsilon))$-DP for all $\epsilon \geq 0$, where*

$$\delta(\epsilon) = \Phi(-\frac{\epsilon}{\mu} + \frac{\mu}{2}) - e^\epsilon \Phi(-\frac{\epsilon}{\mu} - \frac{\mu}{2}).$$

**Theorem 1.2** (Gaussian Mechanism, Theorem 2.7 from of [9]). *Define a randomized algorithm $GM$ that operates on a statistic $\theta$ as $GM(x, \mu) = \theta(x) + \eta$, where $\eta \sim \mathcal{N}(0, sens(\theta)^2/\mu^2)$ and sens is the $l_2$-sensitivity of the statistics $\theta$. Then, $GM$ is $\mu$-GDP.*

Most attempts focused on making gradient boosting private on tree base learners. For example, [5] proposed DPBoost, privatizing gradient-boosted regression trees by finding splits using the exponential mechanism and computing numeric values at leaves using the Laplace mechanism. In a similar fashion, DP-XGBoost was suggested by additionally privatizing the quantile sketching step of XGBoost [6]. DPBoost and DP-XGBoost suffered from low accuracy under strict privacy constraints, as they had to consume privacy budget not only in leaf value computation but also in split finding step (quantile sketching in DP-XGBoost). DP-EBM overcame this by restricting each tree to use only one feature and randomly selecting split points (hence no privacy budget is consumed in split finding) [7].

It showed improved performance compared to DPBoost. However, DP-EBM takes a much longer time to learn a model since it requires learning many more trees.

Absent privacy, gradient boosting on linear models does not improve performance, since linear models are closed under linear combinations, and the base learner can already exactly optimize over this class. But with differential privacy, it is no longer possible to exactly optimize over the base class, so gradient boosting has the potential to give improvements. Moreover, there are private linear learners that make very efficient use of the privacy budget. We adopt a state of the art approach to privately learn linear models, AdaSSP [10], which adds noise to the sufficient statistics for a linear model.

## 2    Differentially Private Gradient Boosting with Linear Models

The flexibility of gradient boosting arises from two choices that we make: (i) a loss function and (ii) a class of base learners. We experiment on three types of loss functions (squared, logistic, and hinge) and fix the base learner class to a private ridge regressor (via AdaSSP). In this section, we describe the loss functions and its gradients (subsection 2.1) and the private base learner class (subsection 2.2).

### 2.1    Loss Functions and Gradients

For binary classification problems, we have a dataset $\mathcal{D}$ of $n$ data points, composed of covariates $x_i \in \mathbb{R}^p$ and labels $y_i \in \{-1, 1\}$ (or $y_i \in \{0, 1\}$ for logistic loss), $\forall i \in [n]$. The final model $f(x)$ takes the covariates $x_i$ as an input and outputs the score $s_i$, which can be translated later to output binary prediction $\hat{y} = \mathbb{1}(s_i > 0) * 2 - 1$ (or $\hat{y} = \mathbb{1}(s_i > 0)$ for $y_i \in \{0, 1\}$ case).

Let $\mathcal{L}$ be the loss function, $T$ be the number of boosting rounds, $f_t$ be the model learned at iteration $t \in [T]$. At $t$-th round of boosting iteration, the goal of Algorithm 1 is to obtain

$$\theta_t = \arg\min_\theta \sum_{i=1}^N \mathcal{L}(y_i, \sum_{k=1}^{t-1} f_k(x_i) + b(x_i; \theta)). \tag{2}$$

The $f_t(x) = b(x; \theta_t)$ can be approximated by steepest gradient descent, where the gradient is taken with respect to the score prediction $s_i$ and evaluated at current score $s_i := \sum_{k=1}^{t-1} f_k(x_i)$. The components of the negative gradient at $t$ can be written as

$$g_{i,t} := -\frac{\partial \mathcal{L}(y_i, s_i)}{\partial s_i}\bigg|_{s_i = \sum_{k=1}^{t-1} f_k(x_i)}. \tag{3}$$

Table. 1 lists the three loss functions we experiment in this paper, and their negative gradients. Note that the squared loss is unbounded, hence we clip the gradient to fall between $(-z, z)$ for some $z \in \mathbb{R}$ (further explained in the next section).

Table 1: Loss functions and gradients, where $\sigma(x) = \frac{1}{(1+e^{-x})}$ and $\mathbb{1}(\cdot)$ is the indicator function.

|  | $\mathcal{L}(y_i, s_i)$ | $g(s_i) = -\partial \mathcal{L}(y_i, s_i)/\partial s_i$ |
|---|---|---|
| Squared | $\frac{1}{2}(y_i - s_i)^2$ | $y_i - s_i$ |
| Logistic | $-y_i \ln(\sigma(s_i)) - (1 - y_i) \ln(1 - \sigma(s_i))$ | $y_i - \sigma(s_i)$ |
| Hinge | $\max(0, 1 - y_i s_i)$ | $\mathbb{1}(1 - y_i s_i > 0)y_i$ |

### 2.2    Private Ridge Regressor as a Base Learner

With the negative gradients $g_{i,t}$ computed as in eq. (3), we fit the next base learner $f_t(x)$ to those gradients by minimizing empirical risk with a ridge regularizer. As we fixed the base learner class to linear models, we may express $f_t(x) = \theta_t^\top x$, and the new model is obtained by

$$\theta_t = \arg\min_\theta \sum_{i=1}^n (\theta^\top x_i - g_{i,t})^2 + \lambda ||\theta||_2^2, \tag{4}$$

where $\lambda \in \mathbb{R}$ is a hyperparameter for ridge regularizer. Note that the objective to be minimized in this step is different from the loss function we chose when computing gradients. Let $X \in \mathbb{R}^{n \times p}$ be the matrix with $x_i$'s in each row and $g_t \in \mathbb{R}^n$ be a vector containing all sample's gradient at $t$ (i.e., $g_{i,t}$). Absent privacy, the above minimization yields

$$\theta_t = (X^\top X + \lambda I)^{-1} X^\top g_t. \tag{5}$$

Using the squared loss function, we can analytically show that $\theta_t = 0 \ \ \forall t > 1$ when $\lambda = 0$, and they are close to zero with small $\lambda$ values. This affirms why gradient boosting was not applied on linear base learners under no privacy constraints.

To meet the privacy constraints, we adopt AdaSSP to privately learn this ridge regressor $\theta_t$ (Algorithm 2 of [10]). Let $\mathcal{X}$ and $\mathcal{Y}$ be the domain of our data covariates and labels, respectively. We define the bound on data domain $||\mathcal{X}|| = \sup_{x \in \mathcal{X}} ||x||$ and $||\mathcal{Y}|| = \sup_{y \in \mathcal{Y}} |y|$. Given the privacy budget $\epsilon, \delta$ to guarantee $(\epsilon, \delta)$-DP, and bounds on the data $||\mathcal{X}||$ and $||\mathcal{Y}||$ for $x_i$ and $g_{i,t}$, respectively, AdaSSP calibrates $(\epsilon, \delta)$-DP to $\mu$-GDP with an appropriate $\mu$, and adds calibrated Gaussian noise to three sufficient statistics: 1) $X^\top X$, 2) $X^\top g_t$, and 3) $\lambda$. The detailed description of AdaSSP algorithm for learning one ridge regressor is deferred to Appendix A.2.

Let $\widehat{X^\top X} = GM(X^\top X, \mu_1)$, $\widehat{X^\top g_t} = GM(X^\top g_t, \mu_2)$, $\widehat{\lambda} = GM(\lambda, \mu_3)$ be the private release of sufficient statistics from a single instantiation of AdaSSP to learn $\theta_t$ and $GM$ is defined in Theorem 1.2. The final model $\widehat{\theta^*}$ can be expressed as

$$\widehat{\theta^\star} = \sum_{t=1}^{T} \widehat{\theta_t} = (\widehat{X^\top X} + \widehat{\lambda} I)^{-1} \sum_{t=1}^{T} \widehat{X^\top g_t} \tag{6}$$

where the initial $g_{i,1} = y_i$. Notice that, for a boosted model, we may call $GM(X^\top X, \mu_1)$ and $GM(\lambda, \mu_3)$ just once and only repeat the second part $GM(X^\top g_t, \mu_2)$ for $T$ rounds, instead of straightforwardly repeating Algorithm 3 for $T$ rounds.

Finally, we suggest BoostedAdaSSP (Algorithm 2), a differentially private gradient boosting algorithm with linear base learners. Algorithm 2 assumes binary classification tasks, and can be generalized to any choice of loss function $\mathcal{L}$. In the second line inside the for loop, we clip the computed gradient to enforce $g_{i,t} \in [-||\mathcal{Y}||, ||\mathcal{Y}||]$, if it is not naturally satisfied. The final output $\theta$ defines the score predictor $f(x) = \theta^\top x$, where the score above 0 means a positive label(+1) and below 0 means a negative label ($-1$ or $0$).

---

**Algorithm 2** BoostedAdaSSP (Data $X, y$, Privacy parameter $\epsilon, \delta$, Split ratio $a, b, c$, Bound on $||\mathcal{X}||, ||\mathcal{Y}||$)

---

**Initialize** $\theta = 0$
Find $\mu$ such that $\mu$-GDP satisfies $(\epsilon, \delta)$-DP. `# Corollary 1.1`
Calibrate $\mu_1, \mu_2, \mu_3$ such that $\mu_1 : \mu_2 : \mu_3 = a : b : c$ and $\mu = \sqrt{\mu_1^2 + \mu_2^2 + \mu_3^2}$.
$\widehat{X^\top X} = GM(X^\top X, \mu_1)$ and $\widehat{\lambda} = GM(\lambda, \mu_3)$ `# instantiate AdaSSP (part 1 & 3)`
$\Gamma = (\widehat{X^\top X} + \widehat{\lambda} I)^{-1}$
**for** $t \in [T]$ **do**
    $s = X\theta$ `# current score prediction`
    $g_t = -\nabla_s \mathcal{L}(y, s)$ `# compute gradient, clip as needed`
    $\theta_t = \Gamma \widehat{X^\top g_t}$, where $\widehat{X^\top g_t} = GM(X^\top g_t, \frac{\mu_2}{\sqrt{T}})$ `# instantiate AdaSSP (part 2)`
    $\theta = \theta + \theta_t$ `# update model`
**end for**
**return** $\theta$
\*$GM(X, \mu)$ denotes a Gaussian mechanism to guarantee $\mu$-GDP for private release of a statistic $X$, and uses the bounds $||\mathcal{X}||$ to compute the sensitivity internally.

---

**Corollary 2.1** (Composition of GDP, Corollary 3.3 of [9]). *The $n$-fold composition of $\mu_i$-GDP mechanisms is $\sqrt{\mu_1^2 + \cdots + \mu_n^2}$-GDP.*

**Theorem 2.2.** *When $||\mathcal{X}|| = \sup_{x \in \mathcal{X}} ||x||$ and $||\mathcal{Y}|| = \sup_{y \in \mathcal{Y}} ||y||$, Algorithm 2 satisfies $\mu$-GDP and $(\epsilon, \delta)$-DP.*

*Proof.* From Corollary 2.1, $\sqrt{\mu_1^2 + T\left(\mu_2/\sqrt{T}\right)^2 + \mu_3^2} = \sqrt{\mu_1^2 + \mu_2^2 + \mu_3^2} = \mu$. The conversion of GDP to DP follows from Corollary 1.1. $\qquad\square$

## 3    Experiments

Algorithm 2 was experimented on five real-world datasets from Kaggle and LIBSVM (details can be found in Appendix A.1) with three choices of loss functions (squared, logistic and hinge), varying number of iterations, and varying choices of epsilon from 0.01 to 10. The privacy parameter $\delta = 10^{-6}$ for all experiments. Lastly, a non-private version was also computed for comparison. The performance of a model is measured by F1 score and AUROC. Note that AUROC may report a good value (close to 1) even when most of the minority class is misclassified. See Appendix A.1 to check how imbalanced each dataset is.

To compare the performance of BoostedAdaSSP against a tree-based private gradient boosting algorithm, we choose DP-EBM [7] which reported the best performance among others[5, 6], and DP-XGBoost [6].

At each boosting round, DP-EBM learns separate trees on individual features, and split points are chosen completely at random (this allows more efficient privacy budgeting). We experimented DP-EBM with four choices of the maximum number of leaves $\{1, 10, 100, 1000\}$. Setting this value to either 100 or 1000 led to similar performance in terms of F1 score, but having fewer splits ran faster, , hence we report results with the number of leaves as 100 in the main paper and defer other results to Appendix A.3. Privacy Accounting this DP-EBM is also through Gaussian DP, which makes it a valuable competitor to ours.

DP-XGBoost [6] learns a differentially private tree each round, and it directly add calibrated noises to aggregate statistics in the XGBoost to make it differentially private. The algorithm uses subsampling to increase the privacy budget per boosting round, and it uses the Approximate DP for privacy accounting. DP-XGBoost may not be a direct comparison to our approach since it doesn't use the same privacy accounting approach as ours, but it is an attempt to make gradient boosting algorithms with trees differentially privacy, so we include their results here. In our experiments, only the sampling ratio is tuned using a pre-defined set of values $\{20\%, 50\%, 80\%\}$, and setting the subsampling ratio to 80% yielded the best performance.

### 3.1    Boosted Linear Models Under Non-Private Regime

Figure 1 shows the training loss of non private gradient boosting, each line corresponding to the choice of loss functions (squared, logistic, and hinge). For the squared loss(green line), we see virtually no improvement over boosting rounds, as mentioned previously in Section 2.2. On the other hand, we observe a decrease in training loss for logistic and hinge losses, when additional boosting rounds are introduced. This is because the gradient of these loss functions are non-linear with respect to the score predictions (whereas it is linear for squared loss).

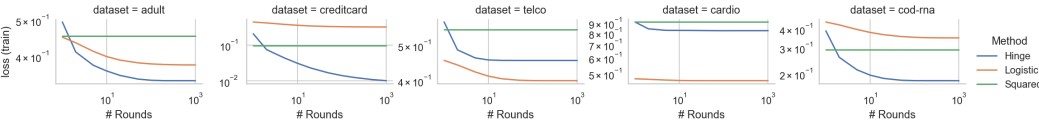

Figure 1: **Training loss versus number of rounds without privacy constraints.** As expected, gradient boosting with the squared loss using a linear model doesn't provide any performance improvement after the first round, however, due to the nonlinearity of the logistic and hinge loss, even with a linear model as the base learner in gradient oosting, more boosting rounds leads more lower training loss.

### 3.2    BoostedAdaSSP (linear base learner) vs. DP-EBM (tree base learner)

Figure 2a and Figure 2b show the performance of private gradient boosting measured by F1 score and AUROC score, respectively. Each line corresponds to BoostedAdaSSP with three choices of loss

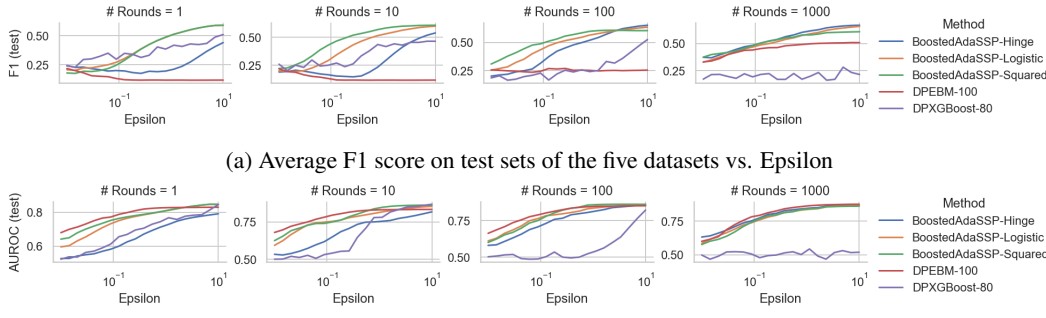

(a) Average F1 score on test sets of the five datasets vs. Epsilon

(b) Average AUROC score on test sets of the five datasets vs. Epsilon

Figure 2: **Averaged scores vs. Epsilon.** On average, our BoostedAdaSSP with squared loss provides higher F1 score than other approaches do, and it also gives comparable AUROC score as our closest competitor — DP-EBM — does. DP-XGBoost, due to its inefficient privacy accounting, underperforms in most cases.

functions and DP-EBM with 100 leaves per tree (red). We observe the improved performance as $\epsilon$ increases ($\delta = 10^{-6}$ is fixed for all experiments) for most cases. This aligns with our expectation that larger privacy budget $\epsilon$ allows less noise to be introduced, leading to a better performance. However, F1 score of DP-EBM in less than or equal to $100$ rounds of boosting behaves counter-intuitively. Since the AUROC score follows our expectation, we may construe this as DP-EBM with small rounds of boosting results in a model that misclassifies majority of the minority class, to the point that the privacy noise sometimes helps correctly classifying the minority labels.

There was no loss functions that outperformed others in all cases—rather, the best-performing loss function (for BoostedAdaSSP) depends on the dataset (detailed results on individual datasets are deferred to Appendix A.5.) Overall, BoostedAdaSSP provides higher F1 scores at most values of total boosting rounds, however, DP-EBM provides slightly better AUROC scores. (Note that the datasets we experiment are mostly imbalanced.)

## 3.3  Effect of Boosting Under Privacy Constraints

Figure 3a and Figure 3b show the performance(y-axis) over boosting rounds(x-axis) at four different privacy levels. The F1 score of BoostedAdaSSP and DP-EBM both improve as the number of boosting rounds increases. DP-EBM requires significantly more number of boosting rounds to yield comparable F1 score to BoostedAdaSSP. To run $1000$ rounds of boosting, DP-EBM takes about $60.9$ seconds to finish, and BoostedAdaSSP takes only about $4.5$ seconds. Overall, we conclude that, restricting the number of boosting rounds to be small (i.e., when we want to limit the time budget), there exists a BoostedAdaSSP with some loss function that is preferrable to DP-EBM for all epsilon values, when we evaluate the performance based on F1 scores.

Additionally for BoostedAdaSSP with squared loss, we compare the ratio between the test set F1 score of a non-boosted model (i.e., with 1 iteration) versus with some rounds of iterations (10, 100, 1000 are plotted). Figure 4 shows that the effect of boosting (measured by the ratio) diminishes as $\epsilon$ goes to infinity as well as when $\epsilon$ goes to zero. This can be explained by conjecturing $\epsilon = \infty$ and $\epsilon = 0$ cases. As the privacy budget $\epsilon$ goes to infinity, our BoostedAdaSSP gradually reduces the amount of noise added to the output, and it eventually becomes similar to the non-private regime. Therefore, we may expect no effect of boosting, the same as in non-private case (see Figure 1). As the privacy budget $\epsilon$ approaches to $0$, we eventually enter the high-privacy regime where the privacy noise dominates the signal. In this case, it is difficult to expect any learning algorithms, let alone additional boosting rounds, to learn anything.

As a result, we observe a *bell curve* shape in the Figure 4, which implies that there exists a sweet spot in terms of the privacy budget $\epsilon$ where the boosting has the maximum impact. However, the sweet spot observed here doesn't necessarily indicates the best F1 score. Same observations are shown in Figure 9 for logistic loss and Figure 10 for hinge loss.

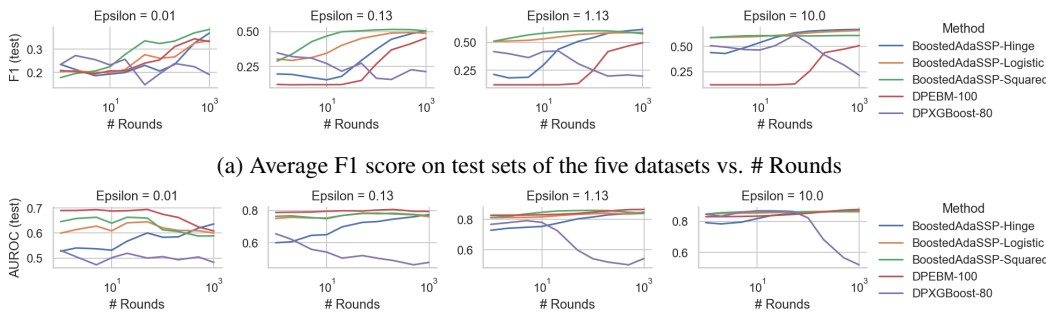

(a) Average F1 score on test sets of the five datasets vs. # Rounds

(b) Average AUROC score on test sets of the five datasets vs. # Rounds

Figure 3: **Averaged scores vs. Epsilon.** At the high privacy regime (small $\epsilon$ value), increasing the number of boosting rounds results in higher F1 score for our approaches with all three loss funcstion and DP-EBM, however, it doesn't improve the AUROC score. The improvement vanishes when we move to the low privacy regime (large $\epsilon$ value). At the same privacy level, DP-XGBoost gives worse performance when more boosting rounds are added.

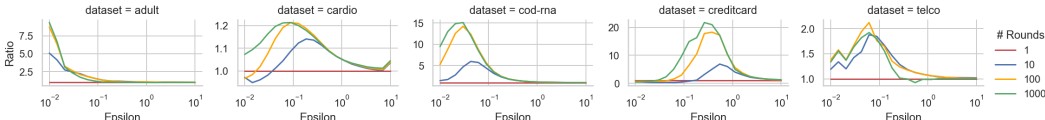

Figure 4: **Effect of DP in gradient boosting with mean squared loss.** It is interesting that the effect of boosting is maximized at a certain privacy level, and the maximum point differs on individual tasks. Thus, it implies that, at the high privacy regime, it might worth running gradient boosting with more rounds for higher performance, but it is not the case in the low privacy regime since the base learner is still linear.

## 4 Conclusion and Future Work

We proposed a differentially private gradient boosting algorithm using linear base learners by adopting AdaSSP to privately train linear models. In each boosting round, the linear model is privately trained to approximate the gradient of loss function at the current score prediction. Without privacy, gradient boosting of a linear model is expected to be the same (OLS) or similar to (ERM with small regularization) a single linear model learned in one-shot. Hence, in practice, gradient boosting is primarily focused on learning with tree base models. However, in the high-privacy regime, BoostedAdaSSP provides a higher F1 score than the state of the art tree-based differentially private gradient boosting algorithm (DP-EBM). BoostedAdaSSP also converges to good performance level with fewer boosting rounds than DP-EBM at a fixed privacy level.

Although the results presented in this paper already seem promising, there are a few ways to further improve the algorithm. One direction could be introducing more hyperparameters to the algorithm. For example, we may introduce a step size $\eta$ to the last step inside the for loop of Algorithm 2. When a weak base learner $\theta_t$ is added to the final model $\theta$, we may multiply $\theta_t$ by $\eta$, as we do in gradient descent algorithms (that being said, Algorithm 2 can be seen as using $\eta = 1$). Also, we may attempt to clip the gradients more aggressively for squared loss and logistic loss as we iterate over boosting rounds, so that we can add less noise when instantiating the second part of AdaSSP.

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

# A  Appendix

## A.1  Datasets

We describe the five datasets used in our evaluation.

Table 2: Datasets used in experiments

| Name | # features ($p$) | # train samples ($n$) | # test samples | % of positive samples in train set |
|---|---|---|---|---|
| cod-rna | 8 | 271617 | 271617 | 33.33% |
| adult | 123 | 32561 | 16281 | 24.08% |
| creditcard | 30 | 227845 | 56962 | 0.17% |
| telco | 46 | 5634 | 1409 | 26.53% |
| cardio | 19 | 56000 | 14000 | 49.96% |

Table 3: Data sources

| Name | Link |
|---|---|
| cod-rna | `https://www.csie.ntu.edu.tw/~cjlin/libsvmtools/datasets/binary.html#cod-rna` |
| adult | `https://www.csie.ntu.edu.tw/~cjlin/libsvmtools/datasets/binary.html#a9a` |
| creditcard | `https://www.kaggle.com/datasets/mlg-ulb/creditcardfraud` |
| telco | `https://www.kaggle.com/datasets/blastchar/telco-customer-churn` |
| cardio | `https://www.kaggle.com/datasets/sulianova/cardiovascular-disease-dataset` |

## A.2  AdaSSP algorithm to learn a single ridge regressor

Let $\widehat{\cdot}$ denote private versions of the corresponding statistics. Then, AdaSSP privately releases the sufficient statistics of ridge regressor as follows.

---

**Algorithm 3** Private Ridge regression via AdaSSP(data $X, y$, calibration ratio $a, b, c$ , Privacy parameter $\epsilon, \delta$, Bound on $||\mathcal{X}||, ||\mathcal{Y}||$)

---

Find $\mu$ such that $\mu$-GDP satisfies $(\epsilon, \delta)$-DP.  # Corollary 1.1

Calibrate $\mu_1, \mu_2, \mu_3$ such that $\mu_1 : \mu_2 : \mu_3 = a : b : c$ and $\mu = \sqrt{\mu_1^2 + \mu_2^2 + \mu_3^2}$.

$\widehat{X^\top X} = GM(X^\top X, \mu_1)$

$\widehat{X^\top g_t} = GM(X^\top g_t, \mu_2)$  # $g_t$ resides within $||\mathcal{Y}||$

$\widehat{\lambda} = GM(\lambda, \mu_3)$

\*$GM(X, \mu)$ denotes a Gaussian mechanism to guarantee $\mu$-GDP for private release of a statistic $X$, and uses the bounds $||\mathcal{X}||$ to compute the sensitivity internally.

---

Algorithm 3 instantiates three Gaussian mechanisms with $\mu_1, \mu_2$, and $\mu_3$ to privately release each sufficient statistic. Hence the composition

$$\widehat{\theta}_t = (\widehat{X^\top X} + \widehat{\lambda} I)^{-1} \widehat{X^\top g_t} \tag{7}$$

is $(\epsilon, \delta)$-DP. Detailed proof is available in Theorem 3 of [10].

## A.3 DP-EBM

Each line corresponds to DP-EBM with $1, 10, 100$ and $1000$ maximum number of leaves per tree. It is easy to see that the performance of DP-EBM plateaus when the maximum number of leaves is $100$, and the performance doesn't improve after that.

We would also like to point out that, the reason that there are missing dots in these plots is due to the fact that the privacy budget allocation in DP-EBM fails to find a solution, and it usually occurs in the high privacy regime.

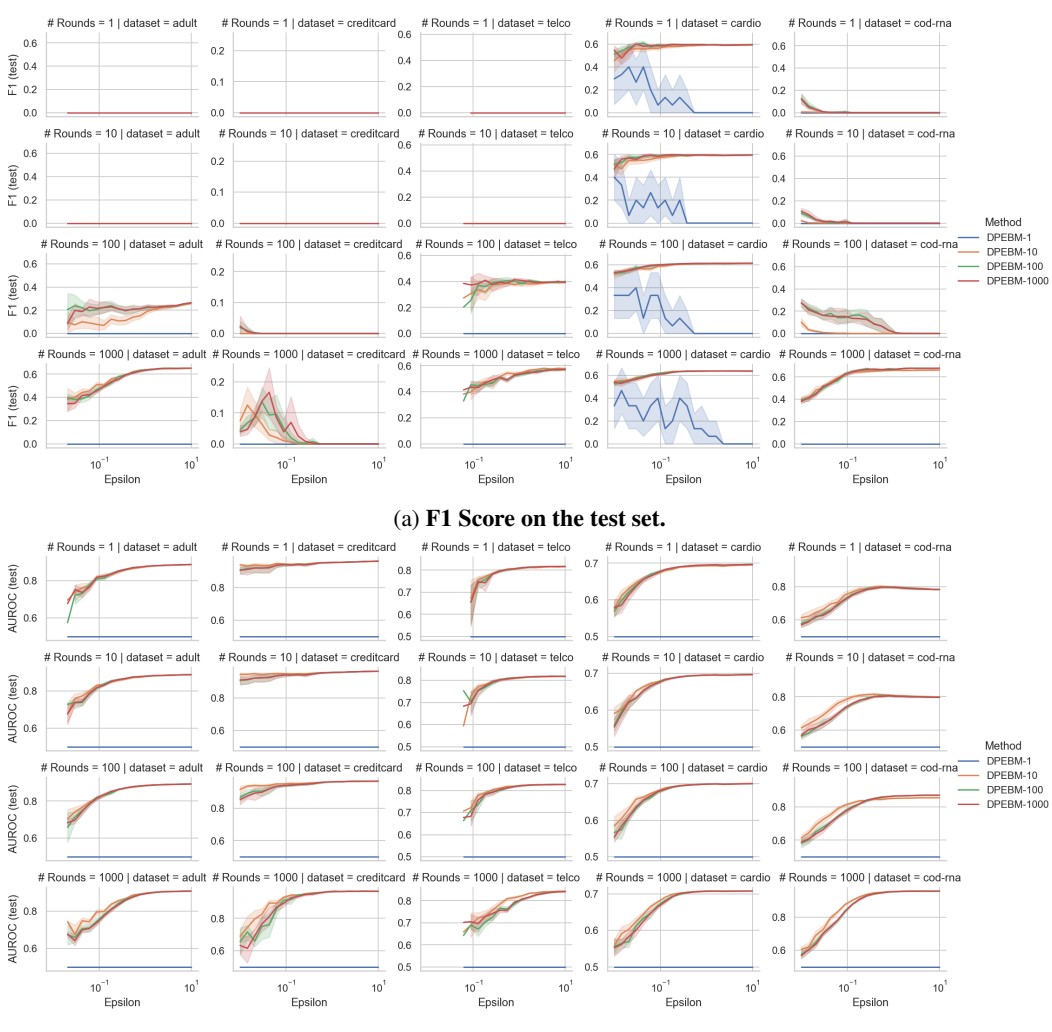

(a) **F1 Score on the test set.**

(b) **AUROC on the test set**

Figure 5: DP-EBM with varying number of leaves per tree.

## A.4  non-private F1, AUROC

In terms of the performance of these non-private boosted linear models on the test set, Fig. 6a and Fig. 6b show F1 scores and AUROC scores, respectively. Apart from the Cardiovascular dataset, increasing the number of rounds improves the F1 score on the test set for logistic and hinge losses only.

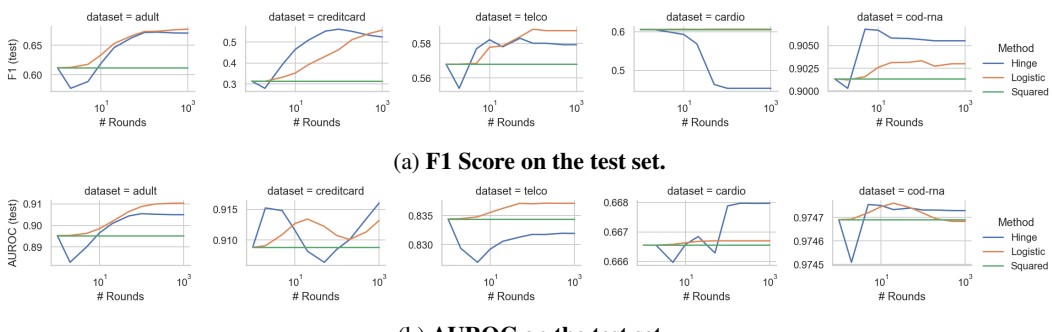

(a) **F1 Score on the test set.**

(b) **AUROC on the test set**

Figure 6: Performance on Non-private Boosted Linear Models on the test set of individual datasets.

## A.5  F1 scores and AUROC scores on the test set of individual datasets

Our main paper presented the averaged F1 score and AUROC score on the test set, and here we present details results on individual datasets. Specifically, the progress of the performance on each dataset is plotted against the privacy level ($\epsilon$ value).

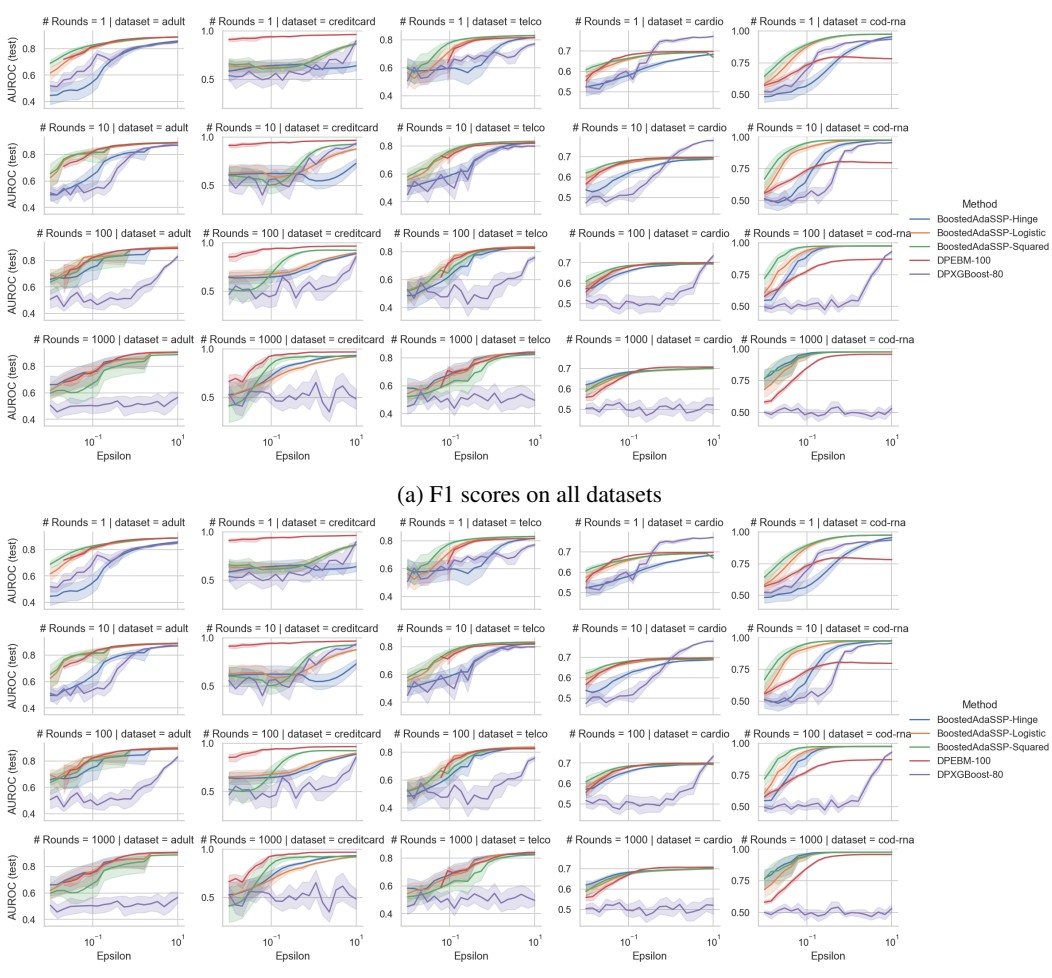

(a) F1 scores on all datasets

(b) AUROC scores on all datasets

Figure 7: Comparisons among BoostedAdaSSP, DP-EBM and DP-XGBoost

## A.6 Effect of Boosting

Here, the progress of the performance on each dataset is plotted against the number of boosting rounds at certain privacy levels ($\epsilon$ values).

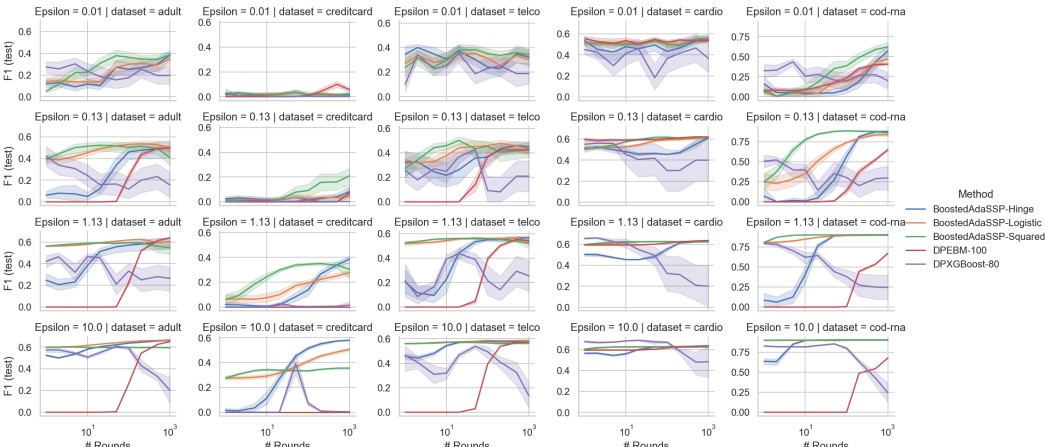

(a) The Effect of Boosting on various methods measured by the F1 score on the test set.

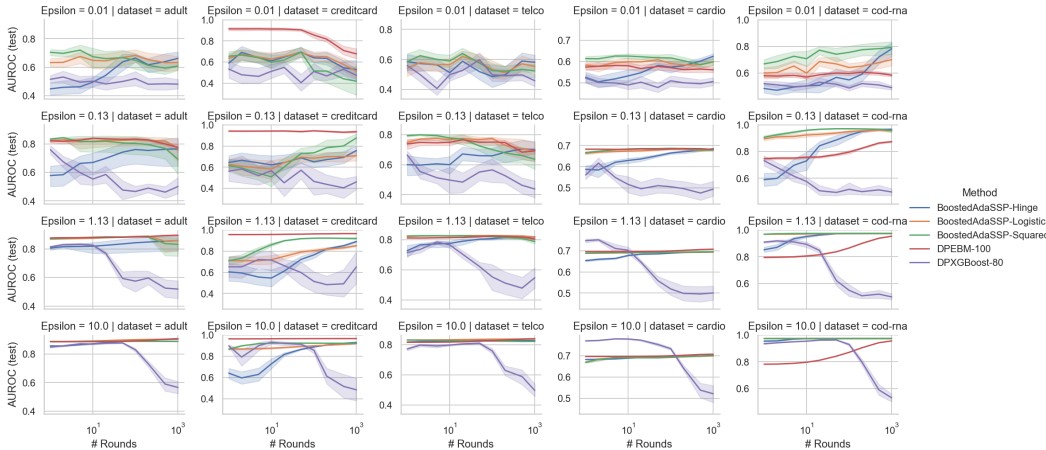

(b) The Effect of Boosting on various methods measured by the AUROC score on the test set

Figure 8: The Effect of Boosting

## A.7 Effect of Differential Privacy

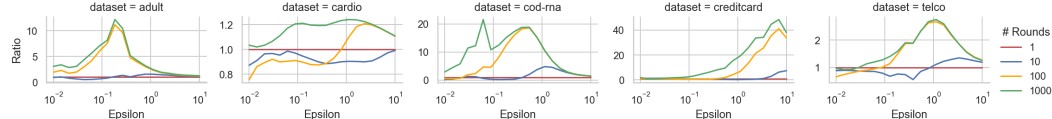

Figure 9: Effect of DP gradient boosting with logistic loss

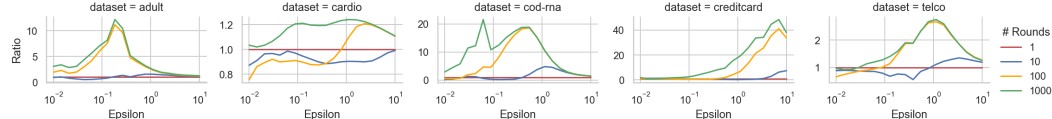

Figure 10: Effect of DP gradient boosting with hinge loss

