# OpenReview forum: "Differentially Private Gradient Boosting on Linear Learners for Tabular Data"
_NeurIPS.cc/2022/Workshop/TSRML — TSRML2022_

### Official Review · Reviewer_iBK8 · 2022-10-19
**An interesting idea to improve DP boosting methods for tabular data, but the experimental evaluation is lacking details.**

**Overall Rating:** 5

**Summary:**

The authors propose a differentially private gradient-boosted classification model based on private linear models that are used as weak learners. The authors argue that private linear learners make efficient use of their privacy budget, and are thus more suitable for learning diff. private GB classifiers compared to tree-based approaches. To this end, the authors use AdaSSP as linear weak learner, which adds noise to the sufficient statistics of the closed-form solution of the linear model. Finally, the authors benchmark their suggested algorithm across 5 tabular data sets and compare this to DP-EBM, a GB approach based on trees.The experimental evaluation suggests that their approach provide more favorable privacy / utility performance relative to DP-EBM.

**Strengths:**

-  Using AdaSSP as linear weak learner, which adds noise to the sufficient statistics of the closed-form solution of the linear model, is a neat idea as these models make efficient use of their privacy budget
- On tabular data gradient-boosting approaches are still state-of-the art  [1] and therefore studying how private GB approaches can be devised is an important problem

----
[1] Vadim Borisov, Tobias Leemann, Kathrin Seßler, Johannes Haug, Martin Pawelczyk, and Gjergji Kasneci. Deep neural networks and tabular data: A survey. arXiv preprint arXiv:2110.01889, 2021.


**Weaknesses:**

I think that the evaluation is lacking sufficient details to conclude whether the proposed method is indeed outperforming the baseline:
- For DP-EBM (the baseline method), the maximum number of leaves is set to 100 across all data sets. The authors argued that the F1 score was then optimized. However, since the optimal hyper parameter value (i.e., 100) lies at the end of the grid (i.e. {2 , 3 , 10 , 100}), it could be that increasing the number of leaves would improve the F1 score.
- The discrepancy between AUC and F1 score results would warrant a more detailed analysis, and averaging scores across data sets can obfuscate things especially when one data set is very highly imbalanced (Credicard, see table 2 in appendix). DP-EBM is not doing well on the F1 score metric, while it does outperform on the AUC metric. For the proposed method, it is the other way around. In summary, looking at the aggregate performances and the fact that one of the data sets is very highly imbalanced makes it difficult to understand the overall performance of the proposed method.

**Overall Recommendation:**

I think that the experimental evaluation is lacking sufficient details to recommend acceptance at this point. I would encourage the authors to fix the weaknesses before submitting this work to a conference or journal.

**Review Confidence:**

3: The reviewer is fairly confident that the evaluation is correct

---

### Official Review · Reviewer_fKCS · 2022-10-22
**Interesting results, but the source of observed performance increase is unclear.**

**Overall Recommendation:** The paper needs work, but might be in…
**Overall Rating:** 6

**Summary:**

The paper introduces a private algorithm for gradient boosting with ridge regression as a base learner. The experimental evaluation finds that this algorithm outperforms private gradient boosted trees.

**Strengths:**

- Unexpected observation that boosting of linear models can be useful in the case of private learning
- Well written paper

**Weaknesses:**

- GDP accountant is approximate and underreports privacy. If the number of compositions differ between the proposed method and DP-EBM, it's better to use another accountant such as Renyi (Moments).
- Comparisons with DP-EBM are certainly well-placed, but it is also interesting to see the performance compared to other DP algorithms, such as DP-Xgboost or DP-SGD for small neural nets.
- An empirical or experimental investigation of why gradient boosting of linear base learners is useful (whereas it is not in non-private setting) would be useful here.

**Review Confidence:**

2: The reviewer is willing to defend the evaluation, but it is quite likely that the reviewer did not understand central parts of the paper

---

### Decision · Program_Chairs · 2022-10-23

**Decision:**

Accept

**Comment:**

Studying the privacy issues on gradient boosted models is a good contribution, but the authors should add missing baselines and more supportive evaluation as suggested by the reviewers in the final revision.